 **eLIFE**

# The metal transporter ZIP13 supplies iron into the secretory pathway in *Drosophila melanogaster*

**Guiran Xiao†, Zhihui Wan†, Qiangwang Fan, Xiaona Tang, Bing Zhou***

State Key Laboratory of Biomembrane and Membrane Biotechnology, School of Life Sciences, Tsinghua University, Beijing, China

**Abstract** The intracellular iron transfer process is not well understood, and the identity of the iron transporter responsible for iron delivery to the secretory compartments remains elusive. In this study, we show *Drosophila* ZIP13 (Slc39a13), a presumed zinc importer, fulfills the iron effluxing role. Interfering with dZIP13 expression causes iron-rescuable iron absorption defect, simultaneous iron increase in the cytosol and decrease in the secretory compartments, failure of ferritin iron loading, and abnormal collagen secretion. dZIP13 expression in *E. coli* confers upon the host iron-dependent growth and iron resistance. Importantly, time-coursed transport assays using an iron isotope indicated a potent iron exporting activity of dZIP13. The identification of dZIP13 as an iron transporter suggests that the spondylocheiro dysplastic form of Ehlers–Danlos syndrome, in which hZIP13 is defective, is likely due to a failure of iron delivery to the secretory compartments. Our results also broaden our knowledge of the scope of defects from iron dyshomeostasis.

**\*For correspondence:**
zhoubing@mail.tsinghua.edu.cn

†These authors contributed equally to this work

**Competing interests:** The authors declare that no competing interests exist.

**Reviewing editor**: Randy Schekman, Howard Hughes Medical Institute, University of California, Berkeley, United States

## Introduction

Iron is critical to the function of a variety of proteins, including hemoglobin and myoglobin, various iron–sulfur proteins, and the electron transport chain. In addition, iron is a necessary component in the secretory pathway. For example, lysyl hydroxylase (LH, also referred to as PLOD) and collagen prolyl 4-hydroxylases (P4Hs) are two post-translational modifying enzymes localized to the lumen of endoplasmic reticulum, which use $Fe^{2+}$ as a cofactor (*Tuderman et al., 1977*; *Pirskanen et al., 1996*). These two enzymes are critical for the synthesis of collagen, a crucial part of the basement membrane, and are formed by a complicated process involving multiple co- or post-translational modifications (*Myllyharju and Kivirikko, 2004*). While the types of collagens and genes found in mammals are very complex, only one type of collagen IV, encoded by two genes *Viking* (*Vkg*) and *Cg25C*, is found in *Drosophila*, and it constitutes a major structural component of basement membranes in the developing fly (*Fessler and Fessler, 1989*).

In adidition to lysyl hydroxylase and prolyl hydroxylase, ferritin is another iron-dependent protein residing in the secretory pathway of *Drosophila*. In constrast to mammalian ferritin, which is predominantly found in the cytosol, *Drosophila* ferritin binds iron in the early secretory compartments and is then secreted into the circulation system (*Mandilaras et al., 2013*; *Tang and Zhou, 2013a*). This process is central for systemic iron supply as well as tissue iron detoxification (*Tang and Zhou, 2013b*). Therefore, in comparison to mammals, it is expected that *Drosophila* will carry a larger amount of iron through this pathway and abnormalities in this process will lead to serious iron deficiency.

Despite considerate interest in iron homeostasis, the metabolic process of iron, particularly its intracellular trafficking, remains poorly characterized. One important question that remains unanswered is how cytoplasmic iron is transferred to the secretory pathway for the iron-dependent proteins found therein.

**eLife digest** Iron is essential for life. Amongst its many important roles, iron is crucial for producing collagen—the protein that provides both strength and elasticity to bones, tendons, ligaments, and skin. Like many other proteins, collagens are produced inside the endoplasmic reticulum—an organelle inside the cell that is enclosed by a membrane that is similar to the plasma membrane that surrounds the cell itself.

Two enzymes that are critical for producing collagen need to bind with iron in order to work correctly. To do this, iron in the cytoplasm of the cell has to cross the membrane that surrounds the endoplasmic reticulum. Small molecules are commonly transported across membranes by proteins called transporters, which tend to work on specific types of ions or molecules. However, researchers did not know the identity of the membrane transporter responsible for moving iron into the secretory pathway—including the endoplasmic reticulum—to bind with the enzymes that produce collagen.

Xiao, Wan et al. have now investigated the function of the transporter ZIP13 in the fruit fly *Drosophila*. This transporter was thought to transport zinc across membranes and into the cytoplasm. Instead, Xiao, Wan et al. found that ZIP13 transports iron out of the cytoplasm and into the endoplasmic reticulum.

Ehlers–Danlos syndrome is a condition that causes individuals to suffer from frequent joint dislocations, bone deformities, and fragile skin as a result of their body producing collagen incorrectly. One form of Ehlers–Danlos syndrome is caused by ZIP13 transporters working incorrectly. However, this was difficult to understand when it was thought that ZIP13 only transports zinc. The discovery that ZIP13 mostly transports iron rather than zinc can explain the link between this transporter and Ehlers–Danlos syndrome: if ZIP13 doesn't work, the collagen-building enzymes cannot get the iron they need to work properly.

Disorders caused by iron deficiencies are normally identified by a few tell-tale symptoms, such as anemia, but these are not seen in Ehlers–Danlos syndrome. Xiao, Wan et al. suggest that iron transport problems could therefore be behind a wider range of diseases and disorders than is currently known.

During the process of using the fruit fly as a model to decipher the functions of zinc transporter dZIP13, we unexpectedly discovered dZIP13 physiologically acts as an iron exporter. dZIP13 belongs to the metal transporter ZIP family (zinc-regulated and iron-regulated transporter proteins or Slc39a); members of this family was reported to generally promote zinc transport from the extracellular space or from intracellular vesicles to the cytoplasm (*Liuzzi et al., 2006*). Our work thus identified the iron transporter required for iron loading in the secretory pathway, which is also the first time a ZIP member has been reported as an iron exporter. This finding implies that failure of iron delivery to the secretory compartments is probably the underlying cause for SCD-EDS (the spondylocheiro dysplastic Ehlers–Danlos syndrome, OMIM #612350), which is due to a mutation in hZIP13 (*Fukada et al., 2008*; *Giunta et al., 2008*). Because SCD-EDS displays none of the classical iron phenotypes, for example, anemia or iron accumulation toxicity, our results also suggest that iron dyshomeostasis is likely involved in a wider spectrum of biological abnormalities than previously thought.

## Results

### The putative *Drosophila* orthologue of human ZIP13 is involved in *Drosophila* dietary iron absorption

BLASTP searches using the amino acid sequences of mammalian ZIP family members revealed that the *Drosophila* genome encodes at least eight putative ZIP proteins (*Lye et al., 2012*; *Qin et al., 2013*). Among them, the protein encoded by *CG7816* shares the highest overall homology with human ZIP13 (45% identity and 58% similarity) (*Figure 1A*), and was named dZIP13 accordingly. In the phylogenetic tree, dZIP13 clusters together with hZIP13 and several other members including catsup (CG10449) and hZIP7 (*Figure 1B*), all belonging to the LIV-1 subfamily of zinc transporters or LZT proteins (*Taylor and Nicholson, 2003*). Several typical features of ZIP family members (*Jeong and Eide, 2013*) are found in dZIP13, including eight transmembrane domains (TM), particularly amphipathic TM4 and

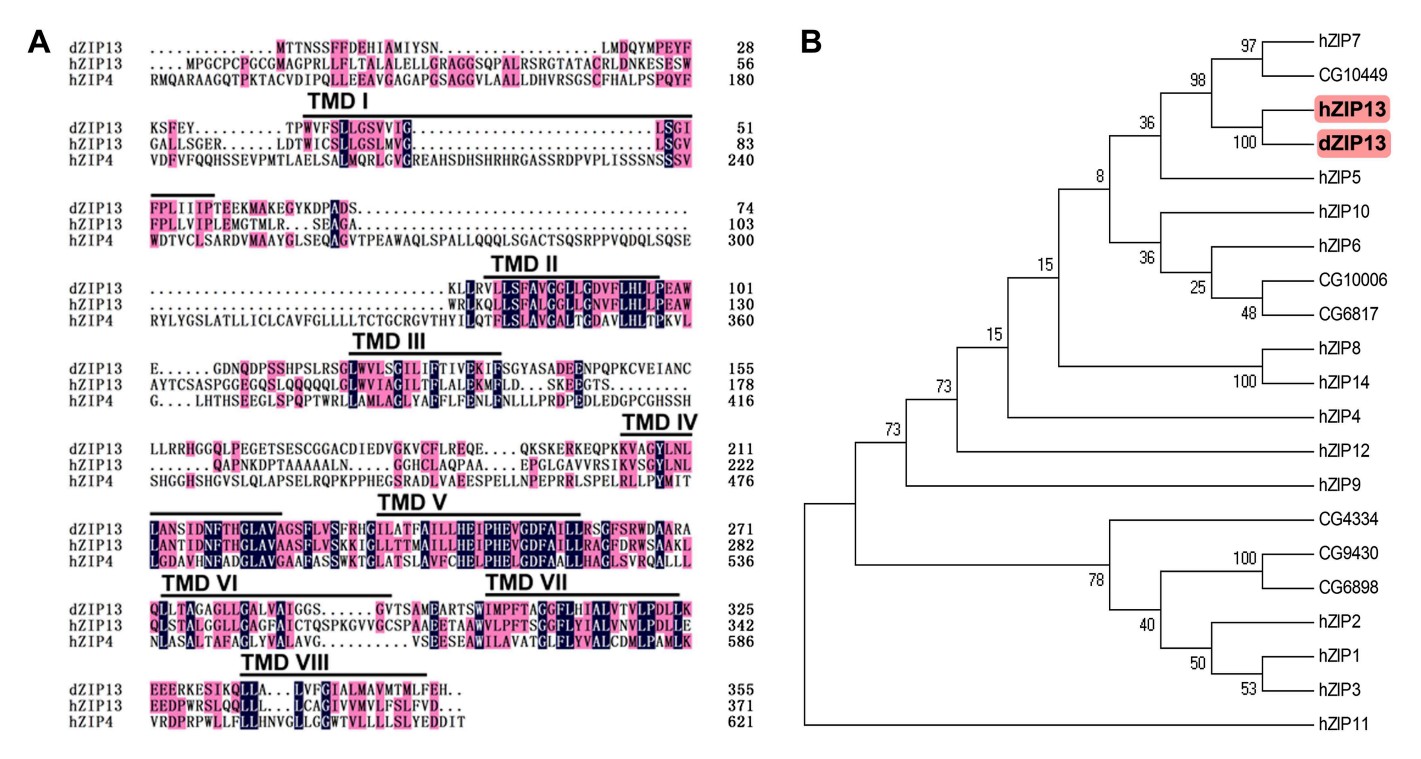

**Figure 1**. Sequence analysis of Drosophila ZIP13. (**A**) Alignment of Drosophila ZIP13 (dZIP13, the top), human ZIP13 (hZIP13, the middle), and human ZIP4 (hZIP4, the bottom) proteins. Amino acid sequences for hZIP13, hZIP4, and dZIP13 (CG7816) were obtained from GenBank and aligned by HMHMM software. Black and pink shadings indicate respectively identical and conservative amino acids. The eight putative transmembrane (TM) regions are underlined and denoted as 'TM I' through 'TM VIII'. (**B**) Phylogenetic tree analysis of human and putative Drosophila ZIP family members. The tree was generated using ClustalX version 1.81 and displayed with TreeView. Bootstrap probabilities for major clusters are shown by percentages. Accession numbers are listed for other Drosophila ZIPs used for the alignment.

TM5, and a predicted extracellular/luminal location of both the amino and carboxyl termini. Notably for dZIP13 and hZIP13, there is only a single His residue in a generally histidine-rich region (2–14 His) between TM3 and TM4. The highly conserved potential metalloprotease His-Glu-*X*-*X*-His (HE*XX*H, where *X* is any amino acid) motif, located within TM5 (*Bin et al., 2011*) of LZT proteins, is also found in dZIP13 (*Figure 1A*).

To analyze the functions of dZIP13 in vivo, transgenic lines of *Drosophila dZIP13* RNAi and overexpression (*dZIP13-RNAi* and *dZIP13-OE*) were generated or obtained, and then tested for dZIP13 expression modulation. The *dZIP13-RNAi* and *dZIP13-OE* indeed efficiently altered expression of *dZIP13* at both the mRNA and protein levels (*Figure 2—figure supplement1* and *Figure 2—figure supplement2*). We initially anticipated dZIP13 as a zinc importer, similar to many other reported ZIP proteins. However, when *dZIP13* was tissue specifically knocked down or over-expressed with *NP3084-Gal4*, a driver specifically expressing the activator Gal4 and thus modulating dZIP13 in the midgut region, the iron levels of the whole body changed dramatically while the amount of zinc level stayed unaltered (*Figure 2A*). Compared to the control, the iron levels in the *dZIP13-RNAi* fly dropped to about 50% of the normal iron content, while iron amount in overexpressing fly increased. This finding was confirmed with different RNAi lines (data not shown), indicating it is not due to off-target effects. The iron effect of dZIP13 suggested to us dZIP13 might directly or indirectly affect dietary iron absorption in the gut.

## Ubiquitous reduction of dZIP13 resulted in developmental arrest that can be rescued by iron supplementation

Under normal dietary conditions, ubiquitous RNAi of *dZIP13* by *daughterless* (*Da-Gal4*) produced developmental arrest in the pupal stage (only about 10% eclosion rate) (*Figure 2B*). Amazingly, the eclosion defect resulting from dZIP13 knockdown could be rescued from ~10% to ~75% simply through

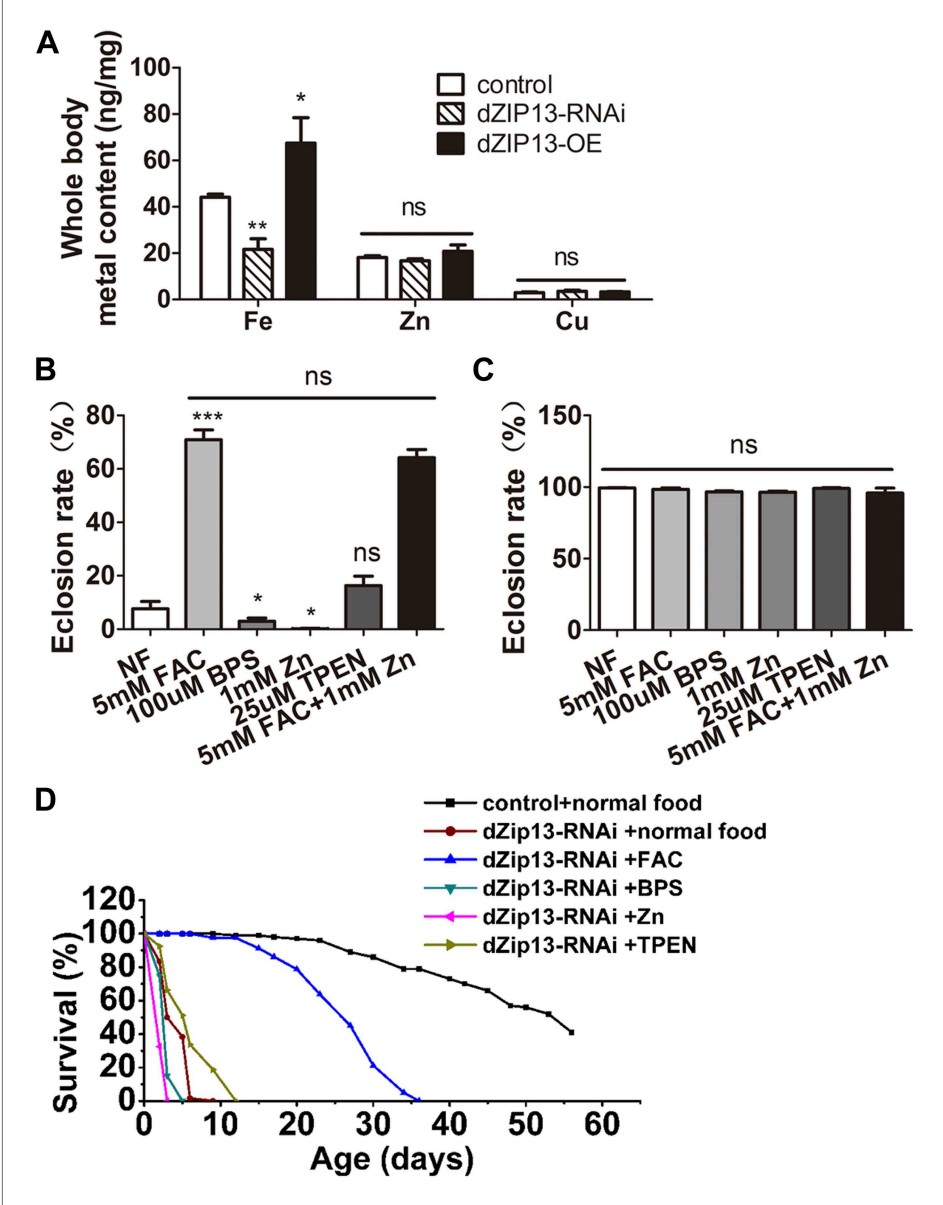

**Figure 2**. *dZIP13-RNAi* flies display iron-rescuable defects. (**A**) Body metal contents when dZIP13 expression was modulated. Shown are flies with modulated *dZIP13* expression in the midgut (*NP3084* as the Gal 4 driver). A significant decrease in the whole body iron content, but not that of zinc or copper, was observed in *dZIP13-RNAi* flies, while *dZIP13* overexpression led to an iron increase. Values represent three independent measurements and are normalized to the dry body weights; data are presented as means ± SEM; *n* = 3 or 6. *p<0.05, **p<0.01; two-tailed Student's *t* test. (**B**) The eclosion rate of ubiquitously-RNA-interferenced-*dZIP13* (*Da > dZIP13-RNAi*) larvae could be rescued by dietary iron supplementation. *Da-Gal4* was crossed to wild-type or *dZIP13-RNAi* flies on juice-agar plates. Newly hatched progeny were transferred to normal food, or food supplemented with ZnCl$_2$, TEPN, FAC, or BPS. Percentages of flies that eclosed to adults were counted; *n* = 6 or 8. (**C**) A control showing that the same amount of zinc, iron, or chelators supplemented in the food had no effect on the eclosion rate of wild-type Drosophila. (**D**) The shortened lifespan of *Da > dZIP13-RNAi* adults was partially rescued by dietary iron supplementation but not zinc. Percentages of flies that eclosed to adults were counted; *n* = 5.

The following figure supplements are available for figure 2:

**Figure supplement 1**. RT-PCR analysis of efficacy of *dZIP13* knockdown or overexpression.

**Figure supplement 2**. Western blot analysis of efficacy of *dZIP13* knockdown or overexpression.

dietary iron supplementation in the form of ferric ammonium citrate (FAC) (*Figure 2B*). Addition of 0.1 mM iron-specific chelator bathophenanthrolinedisulfonic acid disodium (BPS), on the other hand, exacerbated the phenotype as an even lower eclosion rate was observed (*Figure 2B*), indicating abnormal iron absorption indeed plays a critical role in causing the eclosion defect in *dZIP13-RNAi* flies. Zinc, a substrate of many ZIP proteins, also to some extent exacerbate the eclosion defect. When zinc was added to the diet, almost no adult flies could eclose (~0% eclosion rate); however, addition of zinc chelator TPEN could ameliorate the phenotype slightly, although without statistical significance (*Figure 2B*). There have been several reports showing that iron absorption can be competitively inhibited by additional zinc (*Rossander-Hulten et al., 1991*; *Whittaker, 1998*). From that perspective it is possible that zinc addition to the food could exacerbate the iron deficiency phenotype of *dZIP13-RNAi* flies. Simultaneous addition of both zinc and iron could still rescue the eclosion defect very effectively (~65% eclosion rate, statistically insignificant with iron only). The wild type flies' eclosion rate, in contrast, remained the same either in iron supplemented or deficient, zinc supplemented or deficient food (*Figure 2C*). The dramatic rescue by iron instead of zinc suggests that lack of iron is the primary defect in *dZIP13-RNAi* flies, and the zinc effect is minimal and likely secondary. As a result of this unexpected finding, our original focus of dZIP13's function in zinc homeostasis was subsequently switched to investigating its role in iron homeostasis.

Consistent with the results obtained in the eclosion experiments, the lifespan of *dZIP13-RNAi* flies was also prolonged by iron addition in the food (*Figure 2D*): the flies raised on iron supplemented food have a prolonged median lifespan (~25 days) compared with files raised on normal food (~4 days). Addition of zinc to the diet shortened their lifespan (median lifespan of ~1 day). No significant differences of lifespans were found between normal food and BPS or TPEN food.

## dZIP13 knockdown results in iron deficiency of the whole body except the cytosol of the gut cells

The activity of aconitase is often used as a molecular indicator for the availability of iron in the cell (*Haile et al., 1992*; *Suzuki et al., 2005*). There are two types of aconitase in cells, the cytosolic aconitase (c-aconitase) and the mitochondrial aconitase (m-aconitase). Cytosolic aconitase needs to bind iron for its enzymatic activity and is an indicator of the cytosolic iron level (*Haile et al., 1992*; *Tong and Rouault, 2006*). As shown in *Figure 3*, the in-gel aconitase activity assay allowed clear separation of *Drosophila* m-aconitase and c-aconitase, providing a convenient method to evaluate the levels of iron in the two different subcellular compartments (*Tong and Rouault, 2006*).

We examined how the gut itself would be affected when dZIP13 was specifically knocked down in the gut. As shown in *Figure 3A*, *dZIP13-RNAi* larvae exhibited ~35% more c-aconitase activity in the gut as compared to the control larvae while *dZIP13-OE* larvae showed ~20% c-aconitase activity reduction. In contrast to this finding, in the remaining body parts (i.e., the whole body minus the gut) of *dZIP13-RNAi* larvae, c-aconitase activity was significantly reduced when compared to the control larvae, implying a general lack of iron in the body parts other than the gut. Conversely, c-aconitase activity was significantly elevated in *dZIP13-OE,* implying an iron elevation in the other body parts (*Figure 3B*).

Because the labile iron–sulfur cluster of aconitase is also subjected to conditions other than iron levels, such as oxidative stress, we further analyzed how genes involved in iron homeostasis might respond when dZIP13 expression is modulated. We reasoned that because these iron metabolism genes are sensitive to iron availability, their behaviors would also be a good indicator of cytosolic iron levels. Under iron deficiency, ferritin and iron uptake protein *Malvolio (Mvl)* is respectively down- and up-regulated, while under iron surplus, vice versa (*Figure 3C*) (*Missirlis et al., 2007*; *Tang and Zhou, 2013b*). Indeed, when dZIP13 was knocked-down, ferritin was significantly up-regulated while *Mvl* down-regulated, very much akin to the scenario when the larvae were fed with iron-supplemented diet (*Figure 3D*, also see *Figure 4E* for ferritin protein levels). These results indicate dZIP13-RNAi larvae sensed a state of iron-replete condition in the cytosol of their gut cells, consistent with the above aconitase activity results.

To directly quantify the total iron amount, we further performed ICP-MS (inductively coupled plasma-mass spectrometry) analysis of the gut and other body parts (whole body-gut). Both the gut and the rest of the body as a whole exhibited iron reduction after *dZIP13* knockdown and iron increase when *dZIP13* was overexpressed (*Figure 3E* and *Figure 3F*).

Therefore, when *dZIP13* was knocked down in the gut, the rest of the body experienced iron shortage. In the gut, however, although the total iron was lower, iron in the cytosol appeared to not be

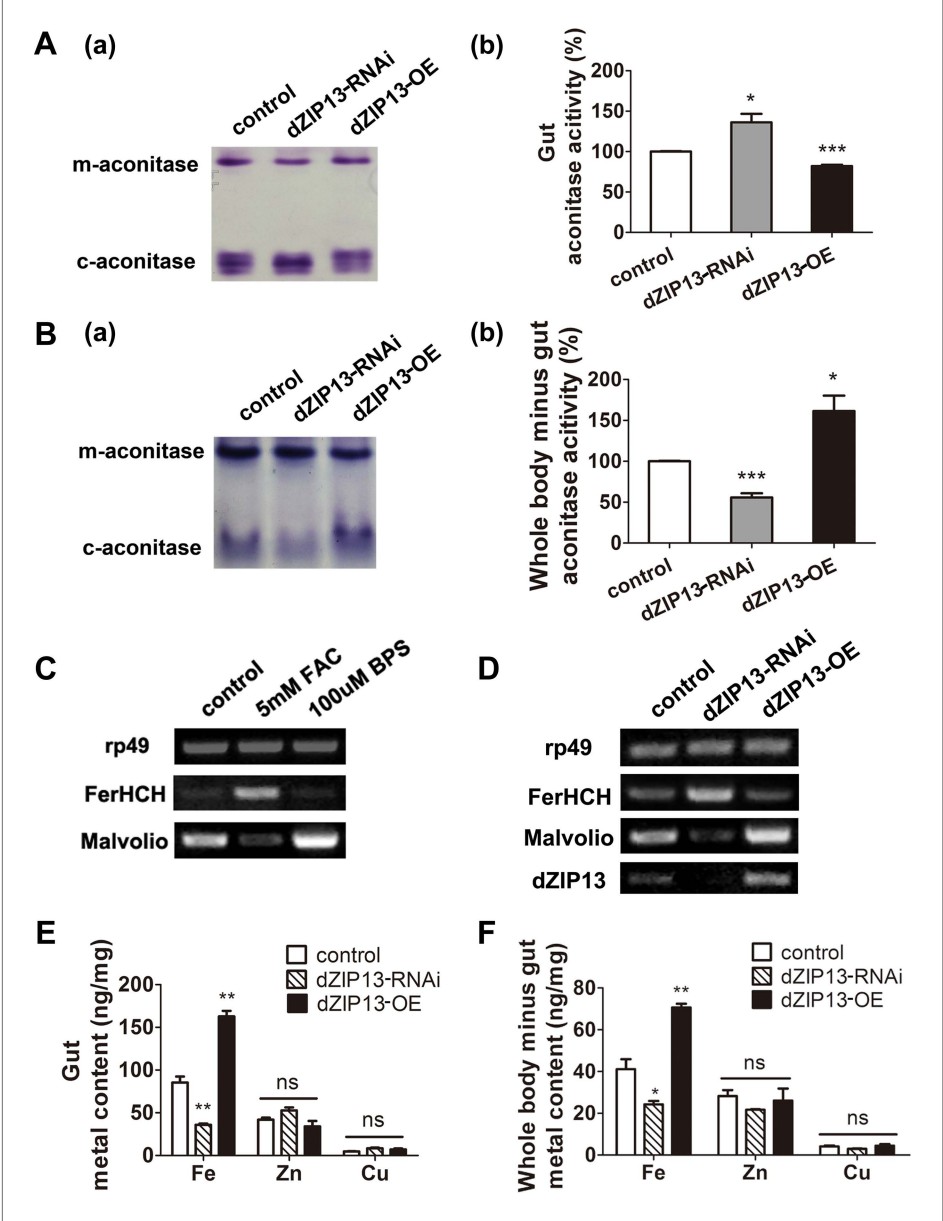

**Figure 3**. *dZIP13* knockdown led to iron deficiency in the body but not the cytosol of gut cells. (**A**) Cytosolic aconitase activity was increased in the gut of *NP3084>dZIP13-RNAi* larvae and decreased in *NP3084>dZIP13-OE* larvae, suggesting respectively iron elevation and iron deficiency in the cytoplasm. Panel (b) was quantitative measurement of (a). Results are presented as mean $\pm$ SEM relative activity; *n* = 3. *p<0.05, **p<0.01, ***p<0.001; two-tailed Student's *t* test. (**B**) Cytosolic aconitase activity was decreased in the whole body minus gut (body parts other than the gut) of *NP3084>dZIP13-RNAi* larvae and increased in *NP3084>dZIP13-OE* larvae. Panel (b) was quantitative measurement of (a). Results are presented as mean $\pm$ SEM relative activity; *n* = 3. *p<0.05, **p<0.01, ***p<0.001; two-tailed Student's *t* test. (**C**) RT-PCR analysis of iron homeostasis genes of normal flies in response to iron changes (feeding with iron or chelator). RNA was made from third instar larvae midguts. *rp49* was used as the loading control. (**D**) RT-PCR analysis of iron homeostasis genes in the midgut of dZIP13 RNAi or OE third instar larvae. Expression is driven by the midgut driver *NP3084*. (**E**) An analysis of metal contents in the gut when dZIP13 expression was modulated. Shown are metal levels from fly larvae with modulated *dZIP13* expression in the midgut (*NP3084* as the Gal 4 driver). A significant decrease in the gut iron, but not zinc or copper, was observed in *dZIP13-RNAi* flies; *dZIP13* overexpression led to an iron increase. Results are presented as mean $\pm$ SEM relative activity; *n* = 3. *p<0.05, **p<0.01, ***p<0.001; two-tailed Student's *t* test. (**F**) An analysis of metal contents in the whole-body-minus-gut parts when dZIP13 expression was modulated in the larval midgut (*NP3084* as the Gal 4

*Figure 3. Continued on next page*

*Figure 3. Continued*

driver). A significant decrease in the whole-body-minus-gut iron, but not zinc or copper, was observed in *dZIP13-RNAi* flies; *dZIP13* overexpression led to an iron increase. Results are presented as mean $\pm$ SEM relative activity; *n* = 3. *p<0.05, **p<0.01, ***p<0.001; two-tailed Student's *t* test.

reduced, suggesting that dZIP13 is involved in iron extrusion from the gut to the body. Because ferritin is the major iron storage protein and is located in the secretory pathway in *Drosophila*, we speculate that when dZIP13 is down-regulated in the gut, iron may not be able to move from the cytosol to the secretory compartments, resulting in iron elevation in the cytosol, a feedback control of iron uptake and an overall reduction of iron in both the gut and the rest of the body.

## dZIP13 is essential to ferric iron loading in the secretory pathway

Because dZIP13 affects iron absorption as shown above, and it is known that the dietary iron absorption is mediated by ferritin in *Drosophila* (*Tang and Zhou, 2013b*), we next investigated the influence of dZIP13 on ferritin iron assimilation. Ferritin, a heteropolymer composed of H and L subunits, can accommodate thousands of iron atoms in its protein shell in the ferric form. Ferritin is thought to be the major cytosolic iron-storage protein in mammalian organisms (*Andrews, 2005*; *Knovich et al., 2009*), while in *Drosophila*, unlike mammals, ferritin sequences of both the H and L chains contain secretion signals, confining them to the secretory pathway and making them abundant in the hemolymph (*Nichol et al., 2002*). A major function of *Drosophila* ferritin is absorption of dietary iron through iron loading and transporting via the secretory pathway supplying iron for the systemic use (*Tang and Zhou, 2013b*). If dZIP13 is responsible for iron loading into the early secretory pathway, we expect it might affect ferritin iron assimilation.

To explore this possibility, we stained the intestines from dZIP13-modulated fly larvae for ferric iron, which is mainly loaded in ferritin. In the iron cell region (a cluster of cells in the middle midgut region where iron absorption occurs) of *dZIP13-RNAi* larvae, we were unable to detect obvious iron signal, whereas a distinct blue iron-staining signal could be observed in the normal fly larvae (*Figure 4A*, *Figure 4—figure supplement1*), suggesting a lack of iron incorporation into ferritin after dZIP13 interference. Conversely, overexpression of dZIP13 led to a stronger staining in the iron cell region, and ectopically in the anterior midgut (*Figure 4A*, *Figure 4—figure supplement1*). This result is reminiscent of the observation that high iron levels, such as 5 mM FAC, can also induce ectopic ferric iron staining in the anterior midgut, suggesting an iron accumulation occurs in the secretory pathway of the gut cells when dZIP13 is overexpressed. Under high levels of dietary iron, iron was only faintly stained in the *dZIP13-RNAi* larvae, whereas staining was much stronger in dZIP13 overexpression larvae (*Figure 4B*). This indicates that in the gut of *dZIP13-RNAi* larvae much less ferric ion is accumulated in ferritin. A native-PAGE staining for ferric iron further confirmed that the intestinal ferritin was indeed poorly loaded with iron in the *dZIP13-RNAi* larvae while the iron loading in the *dZIP13-OE* larvae was mildly elevated (*Figure 4C*, *Figure 4—figure supplement2*).

To exclude the possibility that the poor iron loading observed in *dZIP13-RNAi* larvae was the result of poor ferritin expression, a protein trap line *Fer1HCH*[G188], which expresses an N-terminal GFP-tagged Fer1HCH, was used to indicate the ferritin expression pattern in the gut (*Missirlis et al., 2007*). As shown in *Figure 4D* and reported previously (*Missirlis et al., 2007*), in flies reared with standard food, GFP-tagged ferritin was most prominently expressed in the iron cell region. When dZIP13 expression was knocked down, expression of ferritin was ectopically and significantly induced in the anterior and posterior midguts, and was constitutively expressed in the iron cell region (*Figure 4D*, *Figure 4—figure supplement3*). This expression pattern of ferritin was repeated when flies were administered a diet containing 5 mM ferric ammonium citrate (*Missirlis et al., 2007*), indicating iron is in excess in certain parts of the gut cells when *dZIP13* is RNA interfered, consistent with the above studies in assessing cytosolic iron status of gut cells. Induction of ferritin expression in *dZIP13-RNAi* larvae was confirmed with a Western blot using a ferritin antibody against ferritin. A dramatic increase of ferritin level was seen in *dZIP13-RNAi* larvae, with a decrease observed in the *dZIP13-OE* larvae (*Figure 4E*, *Figure 4—figure supplement4*).

An obvious genetic interaction of dZIP13 and ferritin was detected when dZIP13 was overexpressed in ferritin-RNAi flies. Midgut-specific ferritin RNAi leads to systemic iron deficiency. Ferritin RNAi with

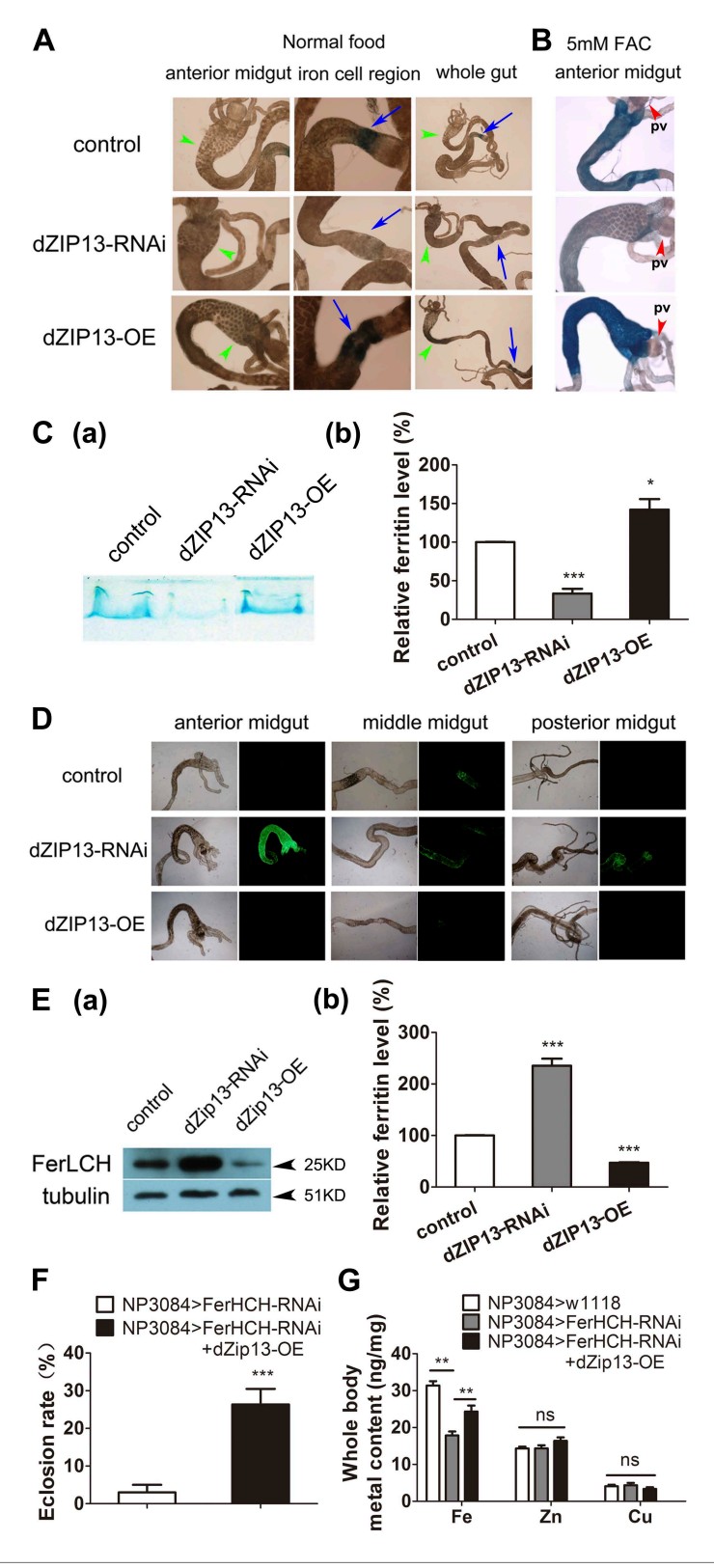

**Figure 4**. *dZIP13* knockdown results in reduced ferritin iron loading in the gut. (**A**) Staining of ferric iron in the larval gut. The staining in the midgut constriction and ectopic ferric staining in the anterior midgut are noted separately by arrows (blue) and arrow heads (green). The anterior midgut of *NP3084>dZIP13-OE* fly larvae deposited

*Figure 4. Continued on next page*

*Figure 4. Continued*

obviously a higher amount of iron than the control. Ferric iron significantly accumulated in the iron cell region of *NP3084>dZIP13-OE* while *NP3084>dZIP13-RNAi* adults showed almost no iron staining. Shown are representative images and in bright and dark fields. More images are shown in ***Figure 4—figure supplement 1***. (**B**) Staining of ferric iron in the anterior midgut of iron-fed larvae. The anterior midgut follows the preceding distinct proventriculus (pv, red arrowheads). (**C**) Staining of ferric iron (bound to ferritin) on native PAGE. Same amounts of total protein extracts from control and *NP3084>dZIP13-RNAi* or *NP3084>dZIP13-OE* larval guts were loaded. The gel was directly stained with Prussian blue staining solution. For an intact gel image, see ***Figure 4—figure supplement 2***. Panel (b) was quantitative measurement of (a). n = 3. *p<0.05, ***p<0.001; two-tailed Student's *t* test. (**D**) Ferritin expression in the gut. An obviously higher amount of ferritin was expressed in the gut of *NP3084>dZIP13-RNAi* larvae. The expression of ferritin was indicated with a protein trap line *Fer1HCH^{G188}*, which tags the endogenous Fer1HCH through an N-terminal GFP fusion. Shown are representative results and more images are shown in ***Figure 4—figure supplement 3***. (**E**) Western blot of ferritin of *NP3084>dZIP13-RNA*i and *NP3084>dZIP13-OE* larvae. Anti-ferritin light chain antibody was used. Tubulin was used as a loading control. For an intact gel image, see ***Figure 4—figure supplement 4***. Panel (b) was quantitative measurement of (a). n = 3. ***p<0.001; two-tailed Student's *t* test. (**F**) Eclosion rescue of ferritin-RNAi by dZIP13. The eclosion rate of gut-specific (*NP3084*) ferritin-RNAi flies was rescued from <5% to ~30% by *dZIP13* overexpression. Newly hatched progeny were transferred to normal food, and percentages of flies that eclosed to adults were counted. *n* = 6. ***p<0.001; two-tailed Student's *t* test. (**G**) Rescue of iron deficiency of ferritin-RNAi by dZIP13. The reduced body iron content in gut-specific ferritin-knockdown larvae was also partially rescued by *dZIP13* overexpression. *n* = 6. **p<0.01; two-tailed Student's *t* test.

The following figure supplements are available for figure 4:

**Figure supplement 1**. Staining of ferric iron in the larval gut.

**Figure supplement 2**. Staining of ferric iron (bound to ferritin) on native PAGE.

**Figure supplement 3**. A significantly higher amount of ferritin was detected in the gut of NP3084>dZIP13-RNAi fly larvae.

**Figure supplement 4**. Western blot of ferritin of *NP3084>dZIP13-RNA*i and *NP3084>dZIP13-OE* larvae.

---

gut-specific driver (*NP3084*) causes decreased total body iron contents, retarded growth, and death of most of the progeny at the larval or pupal stage (***Tang and Zhou, 2013b***). Notably, the eclosion rate of these flies could be rescued from <5–30% through overexpressing *dZIP13* (***Figure 4F***). Consistently, the decreased total body iron content of midgut-specific ferritin RNAi larvae could also be partially rescued by *dZIP13* overexpression (***Figure 4G***).

These results suggest that knockdown of dZIP13 would inhibit iron transport into the secretory pathway to be available to ferritin, reducing iron export from the gut for systemic use, while overexpressing dZIP13 would increase body iron amount by facilitating more iron transportation into the secretion pathway, making less iron available in the cytosol of the gut cells.

## dZIP13 is intracellularly located to ER/Golgi and involved in their iron homeostasis

Locations of hZIP13 have been reported in Golgi and some intracellular vesicles (***Fukada et al., 2008***; ***Jeong et al., 2012***). To examine the intracellular position of dZIP13, a C-terminal myc-tagged dZIP13 was introduced into human intestinal Caco2 cells. Immunofluorescence staining indicated it partially overlapped with ER and Golgi (***Figure 5A***). A similarly tagged (at the C-terminal) dZIP13-EGFP was made and expressed in the flies and shown to be functional. This dZIP13-EGFP is also partially located to the Golgi apparatus (data not shown). To check the intracellular positions of endogenous dZIP13 in the *Drosophila* gut, we generated an antibody against dZIP13. The specificity of the antibody was confirmed by significantly reduced staining in *dZIP13-RNAi* larvae and loss of staining in dZIP13 mutant. Once more we observed an obvious overlapping of dZIP13 with the ER/Golgi apparatus in the *Drosophila* gut staining, along with some staining in additional intracellularlocations (***Figure 5B***). Co-localization of dZIP13 with endosome markers, however, was poor (***Figure 5C***).

As previously discussed, the ferritin-loading experiments indicate iron homeostasis is affected in the secretory pathway after dZIP13 RNA interference. Because dZIP13 is expressed in the secretory

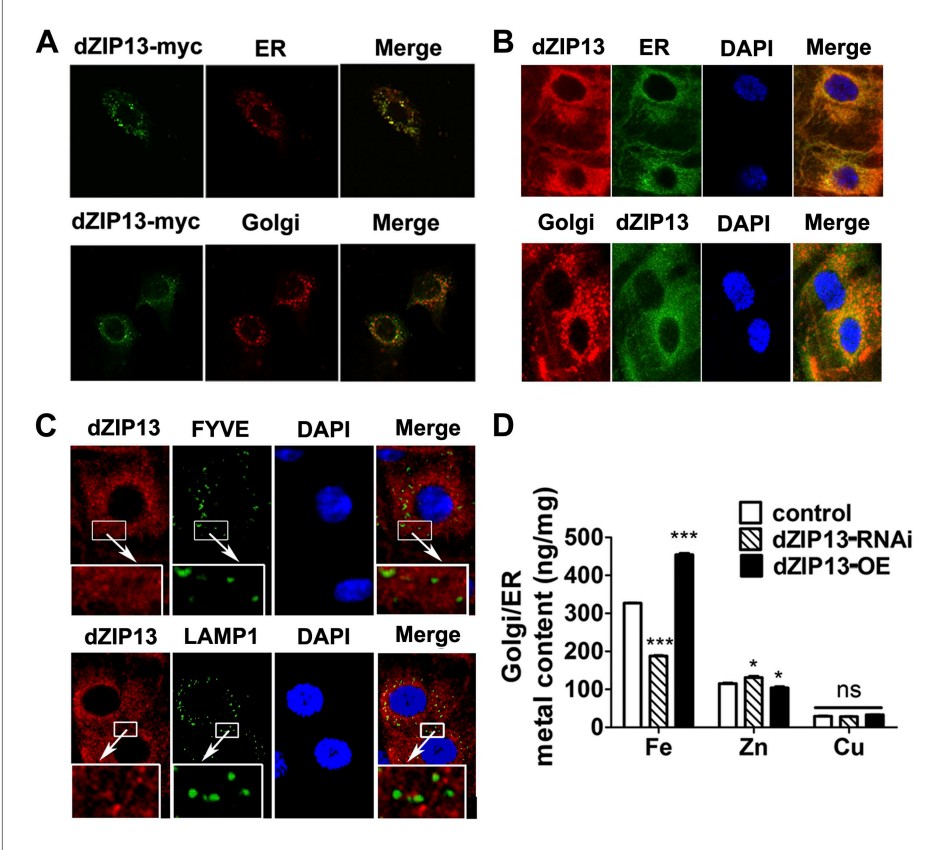

**Figure 5**. dZIP13 is located to ER/Golgi and involved in their iron regulation. (**A**) The localization of dZIP13 in Caco2 cells. dZIP13-myc was detected by myc antibody. This immunofluorescence staining showed that dZIP13 partially co-localizes with the ER/Golgi in Caco2 cells. (**B**) The localization of dZIP13 in Drosophila midgut epithelial cells. dZIP13 was detected directly by dZIP13 antibody. These images indicated dZIP13 partially co-localizes with ER/Golgi in Drosophila gut cells. (**C**) dZip13 does not co-localize well with endosome markers. The localizations of dZIP13 and endosomes in Drosophila midgut epithelial cells were shown. dZIP13 was detected directly with dZIP13 antibody. Lysosome-associated membrane protein 1 (LAMP1) fused with GFP was used to indicate lysosomes and late endosomes; FYVE was used to mark the early endosomes. (**D**) Less iron in ER/Golgi from *dZIP13-RNAi* larvae and more in that of *dZIP13-OE* larvae. Copper was not affected while zinc contents were marginally different. *n* = 3. *p<0.05, ***p<0.001; 2-tailed Student's *t* test.

pathway, we wondered whether iron levels are lower in these compartments when dZIP13 expression is inhibited. Metal content of ER/Golgi was tested and results showed that the iron level of these compartments was indeed reduced in *dZIP13-RNAi* and increased in *dZIP13-OE* larvae (***Figure 5D***). Zinc level was marginally altered in these compartments, but negatively correlated to iron levels (***Figure 5D***). We are not sure, however, if it is a secondary effect of disrupted iron levels or a result of some residual zinc importing activity of dZIP13.

## Heterologous dZIP13 expression confers *E. coli* cells iron-dependent growth and iron resistance

No previous reports have shown a ZIP protein can be a metal exporter. The potential intracellular localization of dZIP13 makes direct assays of its metal transporting activity with intact cells difficult. We reasoned that if dZIP13 indeed functions as a membrane iron exporter, when expressed in *E. coli*, a heterologous platform lacking the sophisticated intracellular membrane system as in eukaryote cells, it might locate to the plasma membrane and facilitate iron cellular export directly outside of the cell. Interestingly, expression of dZIP13 in *E. coli* resulted in very poor growth under normal conditions. Strikingly, growth was partially restored in iron replete conditions whereas zinc had no effect (***Figure 6A***), indicating the lack of growth of *E. coli* with dZIP13 expression was due to intracellular iron

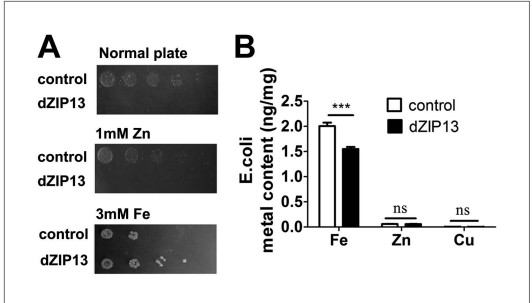

**Figure 6**. Heterologous dZIP13 expression renders the growth of *E. coli* iron-dependent and iron-resistant. (**A**) *E. coli* expressing dZIP13 required iron addition to grow and was more resistant to iron excess. (**B**) Less iron content was detected in *E. coli* expressing dZIP13 while zinc and copper were not much affected. *n* = 3. ***p<0.001; two-tailed Student's *t* test.

deficiency. dZIP13 expression also rendered *E. coli* more resistant to iron stresses compared with the control (*Figure 6A*). Further analysis of cellular iron contents indicated that with iron-rich medium (2 mM iron), expression of dZIP13 reduced the iron level of the cells, while the change of zinc was insignificant (*Figure 6B*). These results suggest that heterologous expression of dZIP13 in *E. coli*, a foreign and arguably a relatively clean system, can reduce intracellular iron availability to cells.

## Radioisotope transporting experiments demonstrate dZIP13 as an iron exporter

Although previous data strongly suggest dZIP13 functions as an iron exporter, they have yet to provide direct proof. One experiment to show a protein as an iron exporter is via a radioactive Fe transport assay. Ferroportin, a plasma membrane iron exporter, for example, has been successfully shown to mediate iron export in *Xenopus* oocytes, by using Fe isotopes and measuring intracellular and extracellular radioactivity levels (*Donovan et al., 2000*). However, as stated above, dZIP13 is located intracellularly and its expression would only directly alter redistribution of iron inside the cell rather than exporting iron out of the cell, as was seen with ferroportin in the oocyte. In order to show dZIP13 indeed acts as an iron exporter, we decided to measure iron effluxing activity of dZIP13 when expressed in *E. coli*, taking advantage of the fact that the only membrane an *E. coli* cell has is its plasma membrane. When dZIP13 is located on the plasma membrane, the radioisotope exporting activity can be directly measured by monitoring the radioactivity in the external buffer. A time-course study of iron transportation in *E. coli* expressing dZIP13 is shown in *Figure 7A*. The amount of iron exported from *E. coli* expressing dZIP13 increased roughly linearly within the tested time and was significantly higher in comparison to the control. Labeling with anti-dZIP13 indicates that dZIP13 was indeed located on the membrane of *E. coli* cells, while control with the empty vector was with minimal signal (*Figure 7B*). These results demonstrate that dZIP13 in *E. coli* is capable of exporting iron out of the cell.

We further isolated ER/Golgi from *Drosophila* larvae and tested their iron transporting activity. In this case, dZIP13 would transport iron into the ER/Golgi apparatus. By testing activities of enzymes for different compartments, we were able to determine that our extract preparations were reasonably pure except for some residual mitochondria (*Figure 7C*, *Figure 7—figure supplement1* and *Figure 7—figure supplement2*). The results shown in *Figure 7D* indicate that the iron uptake of ER/Golgi isolated from *dZIP13-OE Drosophila* larvae was significantly increased whereas that from *dZIP13-RNAi* was decreased. Again, a linear correlation between time and radioactivity was observed. Collectively, these results provide strong direct evidences in support of our hypothesis that dZIP13 is responsible for iron transport from the cytoplasm into the secretory pathway.

## Human ZIP13 can partially substitute dZIP13's functions in *Drosophila*

Bioinformatics analysis indicates that dZIP13 shares the highest homology with human ZIP13 (*Figure 1A*, *Figure 1B*). Because ubiquitous overexpression of dZIP13 resulted in reduced body aconitase activity, we wanted to see whether overexpression of hZIP13 would cause a similar effect in the fly. Indeed, hZIP13 overexpression in *Drosophila* also led to a decrease of overall aconitase activity (*Figure 8A*), suggesting hZIP13 similarly affected iron metabolism in *Drosophila*. Moreover, a survival assay showed that expression of *hZIP13* rescued the eclosion rate of *dZIP13*-RNAi larvae from 40% to 75% (*Figure 8B*), and doubled the lifespan of *dZIP13-RNAi* flies (*Figure 8C*).

Another member of the ZIP family, hZIP7, is closely related to ZIP13 (*Figure 1B*) and has been shown to be a zinc importer located on the ER/Golgi (*Taylor et al., 2004*; *Huang et al., 2005*). Proteins destined for secretion are at least transiently localized to the ER and could theoretically function during their temporary stay in this organelle. If ZIP13 had acted as a zinc importer as other ZIP members, we

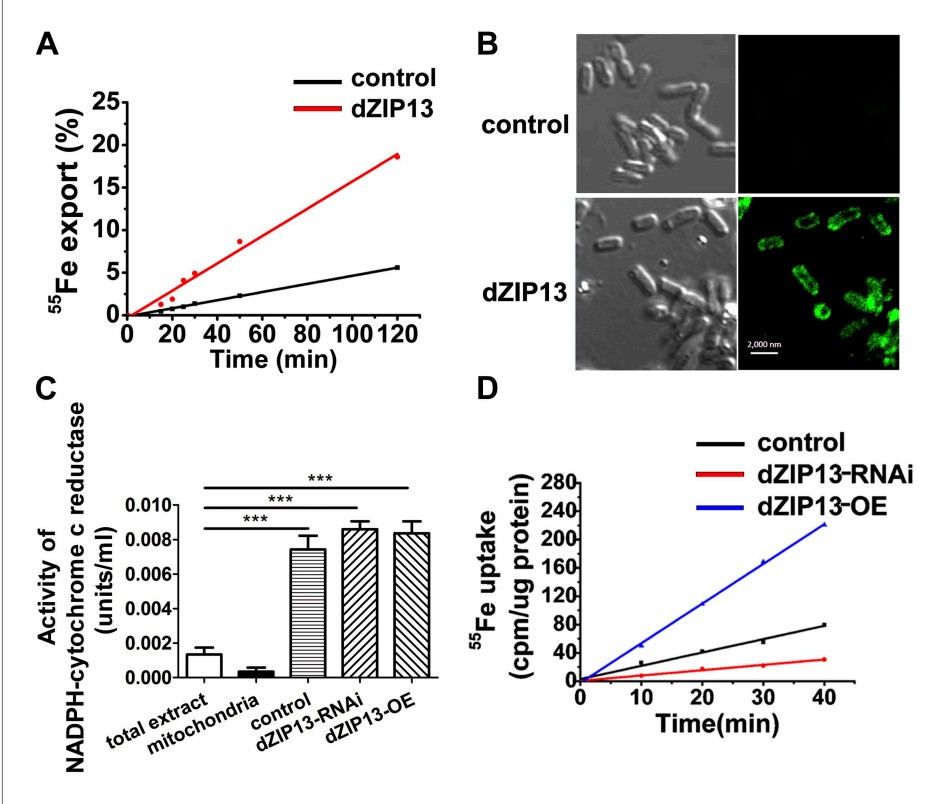

**Figure 7**. Iron radioisotope transport indicates dZIP13 as an iron exporter. (**A**) A time-course of iron released from dZIP13-expressing *E. coli* cells. dZIP13 expression in *E. coli* significantly increased the iron efflux rate than the vector control. (**B**) Imunofluoresence of *E. coli* expressing dZIP13 indicated that at least some dZIP13 was located on the membrane of *E. coli*. (**C**) NADPH-cytochrome c reductase activity assay showed that the ER–Golgi preparation was indeed enriched with a large amount of ER/Golgi. (**D**) A time-course of iron uptake into the ER/Golgi sample. The rate of iron uptake was much higher in *dZIP-OE* and lower in *dZIP13-RNAi* ER/Golgi samples.

The following figure supplements are available for figure 7:

**Figure supplement 1**. Western blot showing that the ER/Golgi samples purified contain both ER and Golgi.

**Figure supplement 2**. Purity of the ER/Golgi samples isolated from Drosophila.

would expect the closely related hZIP7 might also exhibit effects similar to those seen with hZIP13 when expressed in the fly. However, despite similarity to hZIP13, expression of hZIP7 produced neither whole body aconitase activity change, nor rescued the defects of *dZIP13-RNAi* flies (**Figure 8A–C**). In fact, it appeared to have even slightly exacerbated the defects.

hZIP13 mutations cause SCD-EDS, a disease with defective collagen hydroxylation leading to a decrease of collagen cross-linking and secretion. We asked whether dZIP13 knockdown could also affect collagen formation in flies. Type IV collagen, the main constituent of basement membrane, is encoded by two collagen genes *Viking* (*Vkg*) and *Cg25C*, which are expressed in *Drosophila* fat bodies. dPlod is expressed in the collagen-producing cells (**Bunt et al., 2011**), and participates in the assembly and secretion of collagen. We used Vkg-GFP, a GFP protein-trap that identifies the localization of the endogenous Viking protein as a marker of the type IV collagen (**Morin et al., 2001**). Ubiquitous knockdown of dZIP13 caused substantial retention of Vkg-GFP in the fat body cells, while iron supplementation rescued the Vkg-GFP retention (**Figure 8D**). Because iron is an important cofactor/co-substrate of lysyl hydroxylase, a critical enzyme in collagen synthesis (**Figure 8E**) (**Murad et al., 1985**), the observation of iron reduction in ER/Golgi when dZIP13 is inhibited (**Figure 5D**) suggests that the normal functions of lysyl hydroxylase and prolyl hydroxylase might be compromised under iron shortage (see 'Discussion' for more).

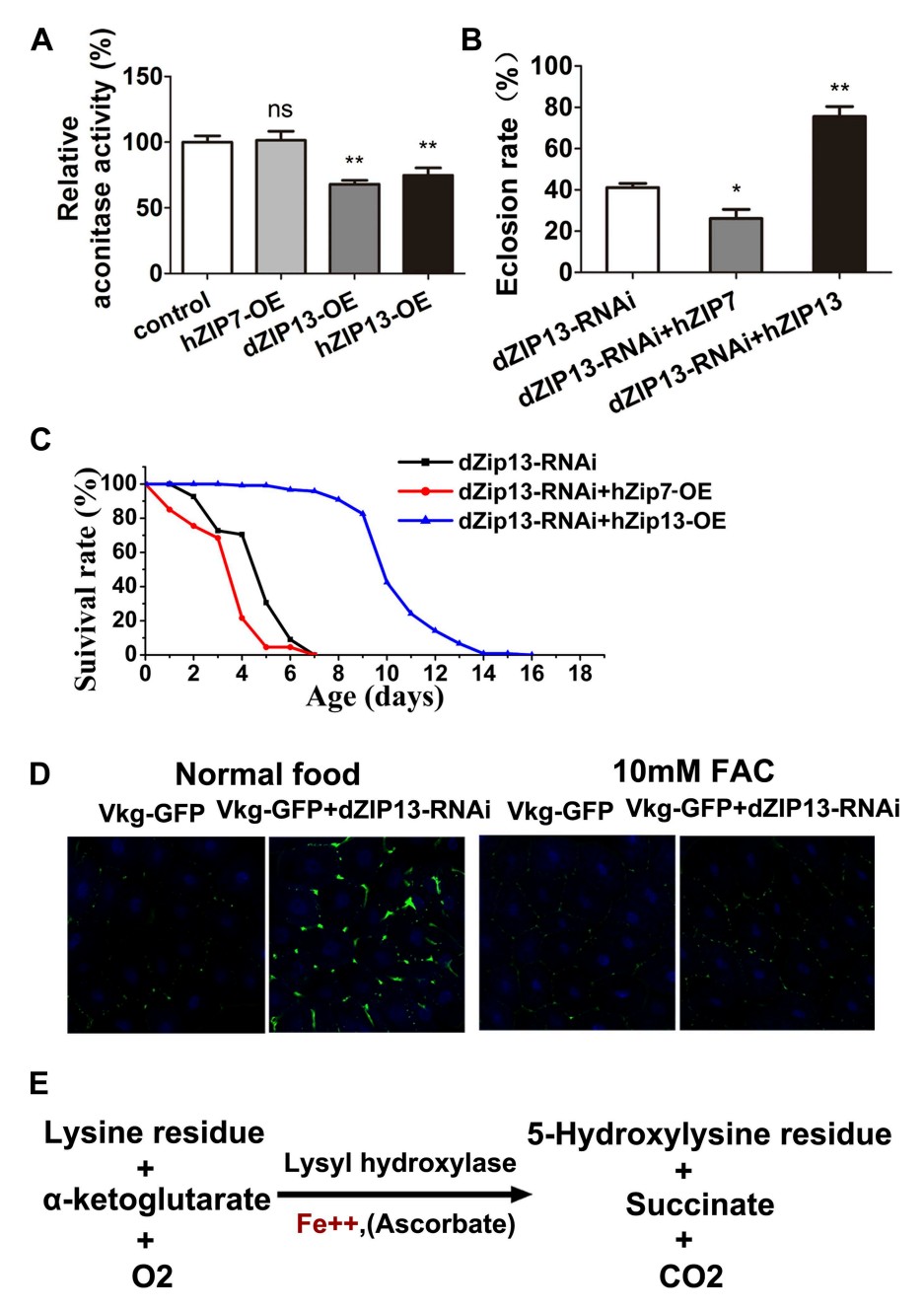

**Figure 8**. dZIP13 is functionally analogous to hZIP13. (**A**) Decreased aconitase activity was observed after either dZIP13 or hZIP13 overexpression, but not a closely related zinc transporter hZIP7. *n* = 6. \*\*p<0.01; two-tailed Student's *t* test. (**B**) The eclosion defect of *dZIP13-RNAi* flies could be significantly rescued by *hZIP13* expression. Eclosion was rescued from ~40% to ~75% by *hZIP13*. *n* = 6. \*p<0.05, \*\*p<0.01; two-tailed Student's *t* test. (**C**) The shortened lifespan of *dZIP13-RNAi* was also rescued by *hZIP13*. (**D**) *dZIP13-RNAi* flies exhibited iron-rescuable collagen defects. Shown are images of fat body cells in VkgG454/+ control larvae and *dZIP13-RNAi* larvae, cultured on normal food and food with 10 mM FAC. Green: Vkg-GFP, blue: DAPI. (**E**) The lysyl hydroxylation reaction. The reaction depends on the presence of ferrous iron.

Taken together, our results indicate dZIP13 is analogous to hZIP13, and is truly the *Drosophila* orthologue of hZIP13. Their functions are fundamentally different from those of the other zinc importers including the closely related ZIP7.

## Discussion

We presented substantial evidence establishing that dZIP13 mediates iron export to the secretory pathway, identifying dZIP13 as the elusive iron transporter in the secretory pathway. Moreover, this is the first time to report that a member of the ZIP transporter family functions as an iron exporter. Previously, certain members of ZIPs, such as ZIP14 (*Liuzzi et al., 2006*) and ZIP8 (*Wang et al., 2012*), have been reported to be capable of transporting iron. However, they are only complex, broad-scope metal-ion importers and their in vivo involvement in iron homeostasis is yet to be established. Our in vivo studies clearly demonstrated that expressional alteration of dZIP13 mainly affects iron instead of zinc metabolism. Modulating dZIP13 expression in the gut cell resulted in opposite iron changes in the cytosol and secretory organelles, and radioisotope iron transport assay further showed that dZIP13 mediates iron exporting, providing compelling evidence in supporting dZIP13 as an iron efflux pump. Future biochemistry/biophysical work is needed to decipher the exact mechanistic detail of the transportation process and evidences thus obtained may further substantiate the conclusion of dZIP13 as an iron exporter.

The unanticipated finding of dZIP13 as an iron exporter was facilitated by the characteristic manner of dietary iron uptake in *Drosophila* gut, which is mediated by secreted ferritin and requires iron loading in the secretory pathway (*Tang and Zhou, 2013b*). The subcellular distributions of ferritin are very different between mammals and insects. In mammals, ferritin is predominantly localized in the cytosol for storing cytosolic iron, and absorbed dietary iron is effluxed from enterocytes through ferroportin (*Vanoaica et al., 2010*). However, in insects such as *Drosophila*, ferritin is mainly secreted, playing a central role in iron absorption by bringing iron out of enterocytes and into the hemolymph through the secretory pathway (*Tang and Zhou, 2013b*). This feature of iron absorption demands higher levels of iron to be pumped into the secretory pathway for systemic use, and facilitated our characterization of dZIP13 as the iron pumper which fulfills this role.

In addition to the iron transport function, we also occasionally observed a minor zinc connection for dZIP13. It is possible that dZIP13 still retains some residual zinc-importing activity. However, in most cases, the zinc effect is physiologically marginal or insignificant. We would argue that the zinc dyshomeostasis is not the primary effect of dZIP13 function loss, because iron but not zinc levels were altered in the body of *dZIP13-RNAi* and *dZIP13-OE larvae*. In addition, iron but not zinc supplementation is able to rescue loss of dZIP13. Finally, no genetic interaction was observed between dZIP13 and other ZIPs in the ER/Golgi. These results argue against the idea that zinc dyshomeostasis is the primary defect in the loss of dZIP13 function, and that iron dyshomeostasis is secondary to a zinc defect. Our other characterizations of dZIP13, as reported in this work, are all consistent with this notion.

Considering the unusual nature of dZIP13 as an iron exporter while other members of the family (SLC39A) so far identified all appear to be importers, it is noteworthy that zinc efflux protein ZnT5 (SLC30A5) can mediate zinc transport in both directions (*Valentine et al., 2007*). It would be interesting to know what sequences or structures endow dZIP13 with this unique property. At this stage we do not yet have clear answers to this question. Amino acid sequence comparisons of ZIP13 with other ZIP family members reveal some distinct differences. The most notabe difference is in transmembrane domain TM4. The TM4 amino acid sequence for all ZIP13s are DNFTHG (*Figure 9*) whereas other closely related ZIPs are HNFTDG, i.e., the highly conserved region in TM4 has a D and H residue swap between ZIP13s and other ZIP members. Our preliminary evidences suggest this variation might be an important factor because when we switched this D and H dZIP13 was no longer functional. Additional experiments are clearly needed to further explore this promising lead.

Mutations in LH family genes have been found in several human diseases including Bruck syndrome (*van der Slot et al., 2003*; *Ha-Vinh et al., 2004*) (OMIM #609220) and Ehlers–Danlos type VI (*Yeowell and Walker, 2000*) (OMIM #225400). Recently mutations in ZIP13, a member of the SLC39A/ Zrt-Irt-like protein (ZIP) family, have been identified as the cause for the spondylocheiro dysplastic Ehlers–Danlos syndrome (SCD-EDS), a form of EDS sharing some similar clinical presentations with EDS VI, which is caused by mutations in PLOD1 gene encoding lysyl hydroxylase (LH1) (*Fukada et al., 2008*; *Giunta et al., 2008*). Because ZIP transporters were generally believed to import zinc into the cytosol from the extracellular milieu or organellar lumen (*Kambe et al., 2006*; *Lichten and Cousins, 2009*; *Jeong and Eide, 2013*), the pathogenesis of SCD-EDS was difficult to understand and various hypotheses were proposed to explain how the zinc importer could affect iron homeostasis (*Fukada et al., 2008*; *Giunta et al., 2008*; *Jeong et al., 2012*). The identification of dZIP13 as the iron exporter

```
1. (224) SKPKTDISPYTPPEKIKTSGYLNLLANCIDNFTHGLAVAGSFLVSRKVGFLTTFAI (279)
2. (188) LREQEQKSKERKEQPKKVAGYLNLLANSIDNFTHGLAVAGSFLVSFRHGILATFAI (243)
3. (199) AQPAAEPGLGAVVRSIKVSGYLNLLANTIDNFTHGLAVAASFLVSKKIGLLTTMAI (254)
4. (300) PVRPQNAEEEKRGLDLRVSGYLNLAADLAHNFTDGLAIGASFRGGRGLGILTTMTV (355)
5. (290) EPSSCTCLKGPKLSEIGTIAWMITLCDALHNFIDGLAIGASCTLSLLQGLSTSIAI (340)
6. (318) SQSACYWLKGVRYSDIGTLAWMITLSDGLHNFIDGLAIGASFTVSVFQGISTSVAI (373)
```

**Figure 9**. Two conserved amino acids D and H in the fourth transmembrane domain of other ZIPs are switched in ZIP13. Shown here are the fourth transmembrane domain and adjacent sequences of ZIP13 and several other ZIPs. 1–3 are ZIP13 sequences from different organisms. 4–6 are several other representative ZIPs. Predicted TM4 segments are underlined. The highlighted two amino acids D and H are conserved in all other closely related ZIPs while their positions are switched in ZIP13s. Numbers in parenthesis indicate the starting/ending positions of the ZIPs shown. 1. ZIP13 (*Danio rerio*); 2. ZIP13 (CG7816) (*Drosophila melanogaster*); 3. ZIP13 (*Homo sapiens*); 4. ZIP7 (*Homo sapien*s); 5. ZIP8 (*Homo sapiens*); 6. ZIP14 (*Homo sapiens*).

for the secretory compartments suggests a more direct interpretation of SCD-EDS: ZIP13 mutations impair iron transport to ER/Golgi, significantly attenuating the activities of iron-requiring enzymes such as lysyl hydroxylase, leading to defective collagen synthesis and accumulation in the secretory pathway.

Several issues are worthy of clarification here. SCD-EDS was identified to be defective in collagen crosslinking from actions of lysyl and prolyl hydroxylases. However, lysyl and prolyl hydroxylase activities from the cellular extracts of patient cells were reported to be normal (*Giunta et al., 2008*). This is likely the result from the enzymatic assays, which were performed in the presence of iron (*Murad et al., 1985*; *Giunta et al., 2008*). It has been shown that externally supplied iron is needed to do these assays (*Tuderman et al., 1977*), suggesting iron is probably not tightly bound to these enzymes. However, the addition of iron would neutralize or mask the original iron-deficienct state of these enzymes in vivo (in the ER/Golgi of SCD-EDS patient cells), and therefore produce 'normal' in vitro enzymatic activities despite defective in vivo hydroxylation. These seemingly contradictory observations, therefore, are very consistent with our conclusion that ZIP13 is an iron exporter that supplies iron to ER/Golgi.

*Jeong et al. (2012)* used mammalian cells to measure metal uptake activity of ZIP13. Because ZIP13 is normally an intracellular protein, the authors claimed that when overexpressed, some ZIP13 was mistargeted to the plasma membrane, making the uptake assay possible. In that experiment, zinc uptake was observed, but iron failed to compete with zinc in the importing assay. However, no exporting activity was examined in that experiment. As reported and discussed previously, sometimes we also observed slight changes in zinc levels, though we are not sure whether this was a result of direct dZIP13 transportation or due to secondary effects of iron dyshomeostasis. It is possible that dZIP13 may still retain some residual zinc importing activity. If this is the case, then the results from Jeong et al. would not be contradictory with our own findings.

While Jeong et al. proposed zinc deficiency in the ER/Golgi, *Giunta et al. (2008)*, *Fukada et al. (2008)*, and *Bin et al. (2011)* proposed zinc accumulation in the ER/Golgi underlies SCD-EDS. According to this theory, zinc accumulation in ER/Golgi can compete with iron and then affect collagen hydroxylation. Indeed, zinc has been reported as an inhibitor of prolyl hydroxylase (*Tuderman et al., 1977*). However in our hands, we saw only a marginal increase of zinc in the ER/Golgi when dZIP13 was knocked-down, but a much more dramatic change in the iron level. Because hZIP7, a zinc importer, cannot rescue dZIP13-RNAi, we think when dZIP13 is knocked-down the primary defect is not zinc accumulation. We suggest iron dyshomeostasis instead of a zinc defect is probably the primary cause contributing to SCD-EDS. Although we cannot exclude zinc's effect, it appears that it is not the primary reason, or at most a subsidiary factor.

Another issue worth pointing out is that none of the classical iron phenotypes are observed in SCD-EDS patients. Iron is involved in heme and Fe-S synthesis, and iron deficiency is typically observed as anemia. In SCD-EDS, it is possible that iron deficiency is localized to the secretory pathway and cytosolic and mitochondrial iron levels remain not much affected. Therefore, a ZIP13 mutation would result

in a type of iron dyshomeostasis that would not present itself with the typical types of symptons (e.g., skin abnormality, bone malformation, and growth retardation) observed with classical iron deficiency. We still do not know what other proteins in the secretory pathway require iron. In addition to the common EDS-like features, SCD-EDS patients do present other phenotypes such as generalized skeletal dysplasia involving the spine and striking clinical abnormalities of the hands (*Giunta et al., 2008*). Are these additional characteristics also a result of collagen defects or due to abnormalities other than the collagen? This question remains unanswered. Nevertheless, our results add another level of complexity in the regulation of iron homeostasis and some of the consequences that arise from its disruption.

## Materials and methods

### Plasmids

Constructs used for transgenic flies include pUAST-dZIP13 and pUAST-Golgi-mRFP. pUAST-dZIP13 was generated by PCR amplification of the coding region of CG7816 from *Drosophila* cDNA and cloning into pUAST using the following primers: pUAST-dZIP13 F: 5′-GGAATTCAGCCGAAAATGACCACGA ACAG-3′, pUAST-dZIP13 R: 5′-ATAAGAATGCGGCCGCCCTAGTGTTCGAATAGCATGGTCATC-3′; pUAST-Golgi-mRFP was generated by PCR amplification of monomeric RFP with rhomboid-1 (Rho1), a Golgi marker protein (*Lee et al., 2001*; *Chen et al., 2006*) and cloning into pUAST. Construct used for transfecting human CHO cells was pIRESneo-dZIP13-myc, constructed in pIRESneo (clontech) by fusing myc in frame to the C terminal of dZIP13 using the following primers: pIRESneo-dZIP13 F: 5′-CTAGTGATATC AGCCGAAAATGACCACGAACAG-3′, pIRESneo-dZIP13 R: 5′-CGGAATTCCTACAGGTCTTCTTCAGA GATCAGTTTCTGTTCGTGTTCGAATAGCATGGTCATC-3′. Construct used in *E. coli* assay was pET28a-dZIP13, which was generated by cloning dZIP13 into pET28a using the following primers: pET28a-dZIP13 F: 5′-ACG CATATG ACCACGAACAGCAGCTTCTTC-3′, pET28a-dZIP13 R: 5′-TAC GAATTC CTAGTGTTCGAATAGCATGGTCATC-3′, and then transfected into BL21 (DE3).

All the constructs were verified by sequencing.

### Fly stocks, culture media, and transgenics

Unless otherwise noted, flies were normally reared on standard cornmeal media at 25°C and third instar larvae were used. The concentrations of supplemented metals or metal chelators used were as follows: 5 mM ferric ammonium citrate (FAC), 0.1 mM bathophenanthrolinedisulfonic acid disodium (BPS) (Sigma, St. Louis, MO, USA), 25 μM N,N,N′,N′-tetrakis (2-pyridylmethyl) ethylenediamine (TPEN) (Sigma, St. Louis, MO, USA), 2 mM $ZnCl_2$ (Beijing Yili Fine Chemicals Ltd. Co., Beijing, China). Transgenic flies for each pUAST construct were generated in $w^{1118}$ background by P-element-mediated transformation. Information about the flies used in this study is listed in *Table 1*.

### Eclosion and longevity assays

To examine the effects of metals or chelators on the eclosion of *dZIP13-RNAi*, *Da-GAL4* homozygous flies were crossed to *dZIP13-RNAi*, and the progeny were reared on food containing different metals or metal chelators. The density of each vial was controlled to ~100 progeny. The total number of emerging adults of each genotype was counted.

Longevity assays was performed as described previously (*Xiao et al., 2013*). 3-day-old adult females were collected. 20 flies were placed in a food vial and each vial was kept at 25°C with 60% humidity under a 12-hr light–dark cycle. Food vials were changed every 2 days, and dead flies were counted at that time. 10 parallel group tests were conducted for each genotype, and the experiments were repeated at least three times. Percentage increases in lifespan were based on comparing the median survivals to the controls.

### Metal content assay

Flies of each genotype were reared on normal food from eggs until late third-instar larval stage. About 25 larvae or 110 guts or 50 whole body minus guts were collected, weighed, and sent for quantitative elemental analysis with inductively coupled plasma-mass spectrometry (ICP-MS) XII (Thermo Electron Corp, Waltham, MA, USA) by the Analysis Center of Tsinghua University (*Wang et al., 2009*; *Xiao et al., 2013*). For ER/Golgi metal content, *Drosophila* ER/Golgi was purified as previous described (*Graham, 2001*). After the protein concentration was determined, ~1.4 mg protein was sent for ICP-MS. For *E. coli* metal content, ~0.33 g (dry weight) *E. coli* was used.

**Table 1.** Drosophila used in this study.

| Drosophila | Descriptions | Origin |
|---|---|---|
| Da-Gal4 (#8641) | Ubiquitous Gal4 | Bloomington Drosophila Stock Center |
| NP3084 (#113094) | Expresses Gal4 in salivary glands, gastric caecae, and whole midgut in third instar larvae | Genetic Resource Center at the Kyoto Institute of Technology (DGRC) |
| Vkg-GFP (#G00454) | Carries a GFP fused to viking | Flytrap |
| Fer1HCHG188/TM3(#G00188) | Carries a GFP fused to Fer1HCH (ferritin 1heavy-chain homolog) | Flytrap |
| dZIP13-RNAi(#1364) | CG7816 RNAi line | Vienna Drosophila RNAi Center |
| dZIP13-OE | CG7816 over-expression line | This study |
| FerHCH-RNAi(#12925) | Fer1HCH RNAi line | Vienna Drosophila RNAi Center |
| hZIP7-OE | Human ZIP7 over-expression line | This study |
| hZIP13-OE | Human ZIP13 over-expression line | This study |
| Golgi marker | Carries a RFP fused to Rho1 | This study |
| ER marker (#ZCL1503) | Carries a GFP fused to PDI | (*Morin et al., 2001*) |
| Early endosome marker (#39695) | Carries a GFP fused to FYVE | Bloomington Drosophila Stock Center |
| Late endosome marker (#42714) | Carries a GFP fused to LAMP | Bloomington Drosophila Stock Center |
| dZIP13 mutant (#18595) | CG7816 mutant line | Bloomington Drosophila Stock Center |

## Aconitase activity assay

Protein was extracted with PBST (137 mM NaCl, 2.7 mM KCl, 10 mM $Na_2HPO_4$, 2 mM $KH_2PO_4$, and 0.1% Triton X-100, PH 7.4) from cells or tissue samples. Protein concentration was measured by the BCA kit (Thermal). ~60 µg protein for the adult gut extract, or ~250 µg protein for the whole larvae extract was added to 700-µl reaction buffer (50 mM $K_2HPO_4$, pH 7.4, containing 30 µM citric acid). The increase of absorbance at 240 nm was monitored for 2 min as the relative aconitase activity.

In-gel aconitase activity assays were performed essentially as described (*Tong and Rouault, 2006*). Gels consisted of a separating gel containing 8% acrylamide (132 mM Tris base, 132 mM borate, 3.6 mM citrate), and a stacking gel containing 4% acrylamide (67 mM Tris base, 67 mM borate, 3.6 mM cit-rate). The running buffer contained 25 mM Tris pH 8.3, 192 mM glycine, and 3.6 mM citrate. Electrophoresis was carried out at 180 V at 4°C. Aconitase activities were assayed by incubating the gel in the dark at 37°C in 100 mM Tris (pH 8.0), 1 mM NADP, 2.5 mM cis-aconitic acid, 5 mM $MgCl_2$, 1.2 mM MTT, 0.3 mM phenazine methosulfate, and 5 U/ml isocitrate dehydrogenase.

## RT-PCR

Total RNA was extracted with TRIzol reagent (Invitrogen, Carlsbad, CA, USA). cDNA was reverse-transcribed from 1 µg total RNA with TransScript Reverse Transcriptase (TransGen Biotech Co., Beijing, China). Semiquantitative RT-PCR was performed using gene-specific primers to amplify partial regions of target genes. RNA isolation and reverse transcription were performed independently for three times, and no less than three PCR experiments were applied to each cDNA sample.

## Western blot analysis

Mouse polyclonal antibodies were raised against recombinant dZIP13-2 protein fragment (MTEEKM AKEGYKDPADSKLLRSGSADEENPQPKCVEIANCLLRRHGGQLPEGETSESCGGACDIEDVGK VCFLREQEQKSKERKEQPKRSGFSRWDAARAQKEEERKESIKQLE). Briefly, the cDNA fragments encoding the cytosolic side of this protein (dZIP13-2) were synthesized and cloned into pTwin1 (NEB) vector. The recombinant protein was expressed in *E. coli* and purified by chitin beads (NEB), and injected into mice for antibody generation (Institute of Genetics and Developmental Biology, Chinese Academy of Sciences, Beijing, China). The antibody was affinity purified and pre-absorbed before use. Anti-Fer2LCH

was as described before (*Tang and Zhou, 2013b*). Anti-tubulin rat monoclonal antibody (ab6160), anti-GM130 rabbit polyclonal antibody (ab30637), and anti-PDI mouse monoclonal antibody (ab2792) were obtained from Abcam (Cambridge, MA, USA). Secondary antibodies include HRP-conjugated goat anti-mouse IgG, goat anti-rabbit IgG and goat anti-rat IgG (Zhongshan Goldenbridge Biotechnology, Beijing, China). For Western blot analysis, fly samples were homogenized in the buffer containing 1% Triton X-100 plus 10% proteinase inhibitor cocktail (Sigma), centrifuged, separated on 10% SDS-PAGE, and transferred to nitrocellulose membranes (Millipore, Watford, UK). Signals were developed with ECL detection kit (Vigorous Biotechnology, Beijing, China).

## Immunohistochemistry and fluorescence microscopy

Caco-2 cells were maintained in MEM (Invitrogen) containing 10% fetal bovine serum (FBS, Gibco BRL, Gaithersburg, MD, USA) and MEM NEAA (Gibco) at 37°C. Cells were transfected with pIRESneo-*dZIP13-myc* and constructs of ER/Golgi markers using lipofectamine (Invitrogen). After 24 hr, cells were fixed, taken pictures after staining for the myc antibody (green) and the Golgi marker (anti-Giantin,1:500) or ER marker (anti-PDI, 1:500). Anti-c-Myc rabbit polyclonal antibody (1:500) (ab51156), anti-Giantin rabbit polyclonal antibody (ab80864) and anti-PDI mouse monoclonal antibody (ab2792) were obtained from Abcam (Cambridge, MA, USA). Secondary antibodies include cy3-conjugated goat anti-mouse and cy3-conjugated goat anti-rabbit IgG (Zhongshan Goldenbridge Biotechnology, Beijing, China).

For dZIP13 staining in *E. coli*, fixation and permeabilization of cells transformed with pET28a were performed as described previously (*Den Blaauwen et al., 2003*). Poly-L-lysine-coated coverslips loaded with fixed cells were washed three times with PBS, and nonspecific binding sites were blocked for 1 hr in PBS containing 1% bovine serum albumin. Coverslips were incubated with anti-dZIP13 antibody (1:100) for 1 hr, washed three times with PBS, and incubated for an additional 1 hr with FITC-conjugated goat anti-mouse IgG (1:1000). The coverslips were then washed three times with PBS, mounted onto glass slides, and taken pictures.

For fly samples, tissues indicated were dissected, fixed, stained, and mounted following standard procedures (*Pastor-Pareja and Xu, 2011*). The following antibodies and dyes were used: mouse anti-dZIP13 (1:100), goat anti-mouse IgG conjugated to FITC or cy3 (1:1000, Zhongshan Goldenbridge Biotechnology) and DAPI (1:5, Beyotime C1005). For DAPI staining, samples were incubated in 50 ng/ml DAPI for 10 min. Slides were mounted with 50% glycerol/PBS. Confocal images were taken with a Zeiss LSM710 Meta confocal microscope.

Fluorescence of *Fer1HCHG188* larvae was examined and recorded with a Nikon ECLIPES 80i microscope attached to a Nikon DXM1200F digital camera (Nikon, Tokyo, Japan).

## Iron staining

Detection of ferric iron in the midgut or in-gel ferric analysis was performed as described previously (*Tang and Zhou, 2013b*).

## Determination of $^{55}$Fe release from *E. coli*

20 ml freshly inoculated *E. coli* was grown to a density of about OD600 = 0.05 in LB containing kanamycin. The cells were incubated with 10 µCi $^{55}$Fe (PerkinElmer, NEZ04300) in the same medium containing 100 µM ascorbate (Sigma) for 1.5 hr. Then 0.5 mM IPTG was added into the medium and incubated for another 1.5 hr. The bacteria were collected at 8000×g for 5 min, washed with ice-cold PBS containing 100 µM EDTA to remove iron non-specifically bound to the cell surface, and twice with ice cold PBS. To measure $^{55}$Fe release, the bacterial cells were then incubated with 2 ml PBS at 37°C, 200 µl medium was sampled at each time point. Supernatants were collected after centrifugation and counted by liquid scintillation (1450 MicroBeta TriLux, PerkinElmer Life Sciences).

$$\% \ ^{55}\text{Fe release} = (\text{cpm in supernatant})/(\text{cpm in time 0}) \times 100\%$$

## Determination of $^{55}$Fe uptake into ER/Golgi

*Drosophila* ER/Golgi purification was performed as previous described (*Graham, 2001*). After determining the protein concentration, the ER/Golgi solution was diluted into 0.5 µg/µl protein in the presence of 0.25 M sucrose. To measure iron uptake, ER/Golgi samples were added with 50 µM ascorbate (Sigma), 50 µM FeCl$_2$, and 10 µCi $^{55}$Fe and incubated at 37°C. 200 µl samples were collected by filters (Millipore, VCWP01300) every 10 min, and washed three times with isotonic solution (Sigma, I3533). The filters were counted by liquid scintillation (1450 MicroBeta TriLux, PerkinElmer Life Sciences).

## Mitochondria purification

The preparation was performed as previously described (*Miwa et al., 2003*). Third instar larvae were collected and homogenized in a buffer containing 250 mM sucrose, 5 mM Tris–HCl, 2 mM EGTA, 1% (wt/vol) bovine serum albumin, pH 7.4 at 4°C. Protein concentration was measured by the BCA kit (Thermal).

## Activity of NADPH-cytochrome c reductase, cytochrome c oxidase and β-N-Acetylglucosaminidase

Activities of NADPH-cytochrome c reductase, cytochrome c oxidase, and β-N-Acetylglucosaminidase were measured by the kit (Sigma #CY0100, # CYTOCOX1, # CS0780, St. Louis, MO, USA) according to the manufacturer's instructions.

## Statistical analysis

Data were analyzed by Student's t-test between groups, and while multiple groups were compared ANOVA was used. Statistical results were presented as means ± SEM. Asterisks indicate critical levels of significance ($p < 0.05$, $p < 0.01$, and $p < 0.001$).

## Acknowledgements

We are grateful to the Bloomington Stock Center, the *Drosophila* Genetic Resource Center at the Kyoto Institute of Technology, Flytrap, and the Vienna *Drosophila* RNAi Center for fly stocks. We thank Dr Yixian Cui for technical assistance.

## Additional information

### Funding

| Funder | Grant reference number | Author |
|---|---|---|
| National Basic Research Program of China | 2013CB910700, 2011CB910900 | Bing Zhou |
| National Natural Science Foundation of China | 31123004 | Bing Zhou |

The funder had no role in study design, data collection and interpretation, or the decision to submit the work for publication.

### Author contributions

GX, ZW, Conception and design, Acquisition of data, Analysis and interpretation of data, Drafting or revising the article; QF, Acquisition of data, Analysis and interpretation of data, Drafting or revising the article; XT, Acquisition of data, Analysis and interpretation of data; BZ, Conception and design, Analysis and interpretation of data, Drafting or revising the article

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
