## [Decision Letter]

Thank you for sending your work entitled “Slc39a13 supplies iron into the secretory pathway in *Drosophila melanogaster*” for consideration at *eLife.* Your article has been favorably evaluated by Randy Schekman (Senior editor) and 3 reviewers.

The Senior editor and the reviewers discussed their comments before we reached this decision, and the Senior editor has assembled the following comments to help you prepare a revised submission.

In this manuscript the authors examine the role of drosophila Zip13 in the transport of iron and zinc. Zip13 is of particular interest because mutations in humans are associated with a sub type of Ehlers-Danlos, a connective tissue disorder. They present strong and convincing evidence that Zip13 is primarily involved in the transport of iron rather than zinc. This is not without precedent as the ZIP family of transporters has been found to include both iron and zinc transporters in several species. The authors also present evidence that ZIP13 localizes to intracellular vesicles and directs iron efflux from the cytosol to the secretory pathway.

The data regarding the role of Zip13 in iron rather than zinc transport is entirely convincing and highly significant. The significance is increased because of the relevance to human disease. The data regarding the role of Zip13 as an iron efflux transporter is less convincing. As this would be a very unusual property for a metal transporter – some family members functioning as importers with one functioning as an efflux pump – the strength of the evidence needed to support this conclusion may be beyond the scope of this manuscript.

Several concerns are:

1) The use of the in-gel aconitase assay as a measure of cytosolic iron. The labile iron-sulfur cluster of cytosolic aconitase is indeed lost in iron-deficient cells, but conditions other than iron deficiency can also lead to cluster disassembly, ie oxidative stress, nitrosative stress, both of which could be occurring here. Furthermore, the in-gels assay is only semi-quantitative at best. Many of the phenotypes associated with the knockdown or overexpression of Zip13 could also be accounted for by loss or gain of cellular iron uptake. In fact, the reduced level of whole cell iron in the gut cells of knockdowns (Figure 3) supports a role in iron uptake rather than efflux.

2) The use of *E. coli* for expression studies. The lipid environment and mechanisms for inserting polytopic proteins into membranes are very different in prokaryotes vs. eukaryotes. Yeast would have been a much better choice for a model cell in this case. Nonetheless, the data support the authors’ conclusions.

3) The lack of an endosomal marker control in the co-localization studies. If Zip13 is really in ER/Golgi, then an endosomal marker would fail to co-localize.

4) The use of a GFP tagged protein without demonstrating that the tag is not cleaved from the Zip13 in cells. This is needed to confirm that the GFP signal corresponds to that of the rest of the Zip13. The diffuse background staining seen in some images suggests this could be a problem.

However, in support of the authors’ conclusions:

1) the data in Figure 7 offer good support of the authors' hypothesis that Zip13 is mediating efflux from an intracellular compartment.

2) The data regarding the rescue of fly Zip13 knockdowns with human Zip13 are convincing.

3) The use of the Viking-GFP trap is also highly suggestive that the flies have a defect in collagen production.

In summary, the data suggesting Zip13 is involved in iron transport are strong and significant. The data indicating it is an efflux pump are weaker and not entirely convincing. It would be reasonable to acknowledge the limitations of the efflux data and present them as one possible interpretation of preliminary data, but confirmation would require additional evidence.

---

## [Author Response]

*[…] The data regarding the role of Zip13 in iron rather than zinc transport is entirely convincing and highly significant. The significance is increased because of the relevance to human disease. The data regarding the role of Zip13 as an iron efflux transporter is less convincing. As this would be a very unusual property for a metal transporter – some family members functioning as importers with one functioning as an efflux pump – the strength of the evidence needed to support this conclusion may be beyond the scope of this manuscript*.

*Several concerns are*:

*1) The use of the in-gel aconitase assay as a measure of cytosolic iron. The labile iron-sulfur cluster of cytosolic aconitase is indeed lost in iron-deficient cells, but conditions other than iron deficiency can also lead to cluster disassembly, ie oxidative stress, nitrosative stress, both of which could be occurring here. Furthermore, the in-gels assay is only semi-quantitative at best. Many of the phenotypes associated with the knockdown or overexpression of Zip13 could also be accounted for by loss or gain of cellular iron uptake. In fact, the reduced level of whole cell iron in the gut cells of knockdowns (*Figure 3*) supports a role in iron uptake rather than efflux*.

Yes, aconitase activity can also be subjected to oxidative stress. To further corroborate our statement that Zip13 knockdown in the gut indeed led to cytosolic iron increase and Zip13 overexpression cytosolic iron decrease, we examined how genes involved in iron homeostasis would behave in these cases (i.e., do they behave as in iron-rich or iron-poor condition?). Ferritin and iron uptaker Malvolio (fly NRAMP homologue) were chosen. Under iron-replete conditions gut ferritin is induced in both the transcript and protein level, while Malvolio is repressed, and vice versa. We then analyzed ZIP13-RNAi and ZIP13-OE flies. ZIP13-RNAi saw a dramatic increase of ferritin and a depression of Malvolio, exactly coinciding with what has happened under iron replete situation (Newly added data shown in Figure 3, and corresponding text added in the Results). So physiologically speaking, when dZIP13 is knocked-down, gut cells “feel” an iron surplus state, and respond by dramatically increased ferritin expression and reduced expression of iron uptaker Malvolio. These are indirect lines of evidence, but constitute fairly strong indirect lines of evidence together with the aconitase assay in suggesting gut cytosolic iron is reduced when dZIP13 is overexpressed and increased when knockdown.

Contrary to the cytosolic state, that of the secretory compartment is opposite in these cases, i.e., in ZIP13-RNAi and ZIP13-OE flies, iron in the secretory compartments would be reduced and increased respectively. The reason that overall iron reduction is observed in dZIP13-RNAi gut is mostly likely that in fly, ferritin, to which most iron is bound, is in the secretory pathway (unlike mammalians most ferritin is in the cytosol). Indeed, ferric staining highly depends on the presence of ferritin. In other words, the lower total cellular iron in dZIP13-RNAi gut is likely because that gut cells are in a state of iron surplus in the cytosol and a feedback control of iron uptake is thus exerted leading to a reduction of total cellular iron level (cytosolic iron + secretory iron).

The similar expressional change patterns of Mvl and ferritin through dZIP13 RNAi or iron feeding argue against the suggestion that dZIP13 might be involved in iron uptake because if dZIP13 were an iron uptaker, dZIP13-RNAi gut cells would have responded as if lack of iron. Instead, they responded as if they were experiencing an iron overdose.

*2) The use of E. coli for expression studies. The lipid environment and mechanisms for inserting polytopic proteins into membranes are very different in prokaryotes vs. eukaryotes. Yeast would have been a much better choice for a model cell in this case. Nonetheless, the data support the authors’ conclusions*.

Iron transport assay has normally to be performed with surface-resident protein, by measuring extracellular and intracellular radioactivities. For an intracellular membrane protein such as dZIP13 this would be hard to do technically, unless with purified organelle. In other words, expression in yeast would most likely result in intracellular residence of dZIP13, in which case direct iron transport assay with whole cells would be hard to do.

E. coli has no intracellular membrane. So as long as the protein is located to the membrane it will be in the plasma membrane and enable direct measurement of radioactive iron transport. We mentioned this rationale in the original text under the section “Radioisotope transporting experiments demonstrate dZIP13 as an iron exporter”.

We also purified fly ER/Golgi organelle and measured iron transport. The result showed that indeed dZIP13 mediates iron transport to ER/Golgi.

*3) The lack of an endosomal marker control in the co-localization studies. If Zip13 is really in ER/Golgi, then an endosomal marker would fail to co-localize*.

We added two endosomal markers, an early endosomal marker FYVE (Bloomington #39695) and a later endosomal marker LAMP (Bloomington #42714). The results showed that dZIP13 does not co-localize well, if at all, with endosomal markers (new Figure 5).

4) The use of a GFP tagged protein without demonstrating that the tag is not cleaved from the Zip13 in cells. This is needed to confirm that the GFP signal corresponds to that of the rest of the Zip13. The diffuse background staining seen in some images suggests this could be a problem.

Following this advice, Anti-GFP antibody was used to analyze protein extracts from Da>EGFP (as a control) and Da>dZIP13-EGFP 3rd larvae. The results imply that the EGFP tag is not cleaved from dZIP13 protein in vivo (see Figure 10), indicating the original signal seen should be largely from dZIP13-EGFP. The original diffuse background signal is likely due to the fact that the picture was not a cofocal image, and was taken under a Nikon ECLIPES 80i microscope.Author response image 1.Western blot of dZIP13-EGFP larval extract for EGFP to asses EGFP fusion protein stability. Anti-GFP antibody was used. The GFP lane is the extract from Da>GFP flies (4ug). The control lane is normal fly sample (no EGFP expression) (80ug). Little free EGFP was detected in dZIP13-EGFP sample (80ug). An extra band below that of dZIP13-EGFP is likely a background signal because it is also detected in normal fly sample (control lane).

Because of the lesser quality of this picture, its partial redundancy with Figure 5 (which respectively show a myc fusion and dZIP13 antibody staining), and likely misunderstanding arising from it, we decided to remove this particular picture from the current revision.

However, in support of the authors’ conclusions:

*1) the data in*
Figure 7
*offer good support of the authors' hypothesis that Zip13 is mediating efflux from an intracellular compartment*.

*2) The data regarding the rescue of fly Zip13 knockdowns with human Zip13 are convincing*.

*3) The use of the Viking-GFP trap is also highly suggestive that the flies have a defect in collagen production*.

In summary, the data suggesting Zip13 is involved in iron transport are strong and significant. The data indicating it is an efflux pump are weaker and not entirely convincing. It would be reasonable to acknowledge the limitations of the efflux data and present them as one possible interpretation of preliminary data, but confirmation would require additional evidence.

After the revisions, we consider the evidence supporting dZIP13 as iron efflux pump are reasonably or fairly strong. The main supporting evidence, among other evidence, follows:

A) direct evidence is presented to show that when dZIP13 expression is knocked-down, the ER/Golgi lacks iron, while strong indirect evidence is provided (additional evidence provided in this revision as described above) to show cytosol is replete with iron in this case, and vice versa.

B) radioisotope iron transport assays with E. coli and ER/Golgi indicate directly dZIP13 as iron efflux pump.

These would be considered very strong evidence for showing a protein as an iron effluxer. However, as also pointed out by the editor/reviewers, it is very unusual when other reported cases of the family members (ZIP or SLC39A) are all importers and this one turns out as an exporter. Having said this, it is also noteworthy that zinc efflux protein ZnT5 (SLC30A5) can mediate zinc transport in both directions (40). In the revision, we added these comments in the Discussion.

At the end of the first paragraph:

“Our studies clearly demonstrated that expressional alteration of dZIP13 mainly affects iron instead of zinc metabolism. Modulating dZIP13 expression in the gut cell resulted opposite iron change in the cytosol and secretory organelles, and radioisotope iron transport assay further showed that dZIP13 mediates iron exporting, providing compelling evidence in supporting dZIP13 as an iron efflux pump. Future biochemistry/biophysical work is needed to decipher the exact mechanistic detail of the transportation process and evidence thus obtained may further substantiate the conclusion of dZIP13 as an iron effluxing protein.”

And at the beginning of fourth paragraph:

“Considering the unusual nature of dZIP13 as an iron exporter while other members of the family (SLC39A) so far identified all appear to be importers, it is noteworthy that zinc efflux protein ZnT5 (SLC30A5) can mediates zinc transport in both directions (40).”